# Changes in Cultivated Land Area and Associated Soil and SOC Losses in Northeastern China: The Role of Land Use Policies

**DOI:** 10.3390/ijerph182111314

**Published:** 2021-10-28

**Authors:** Haiyan Fang

**Affiliations:** 1Key Laboratory of Water Cycle and Related Land Surface Processes, Institute of Geographic Sciences and Natural Resources Research, Chinese Academy of Sciences, Beijing 100101, China; fanghy@igsnrr.ac.cn; 2College of Resources and Environment, University of Chinese Academy of Sciences, Beijing 100049, China

**Keywords:** cultivated land, soil erosion, SOC losses, land use policy, black soil region

## Abstract

Land use policy is the driving factor influencing land use; however, little research has been conducted to identify the role of agricultural policy in influencing land cultivation and associated soil and soil organic carbon (SOC) losses. The aims of this study were to explore temporal changes in cultivated land, soil erosion, and SOC loss and to identify the role of land use policy. The present study was conducted using the revised universal soil loss equation by integrating remote sensing images from 1980, 1990, 2000, and 2017. The study found that cultivated land areas increased from 275.11 thousand km^2^ in 1980, to 300.03 thousand km^2^ in 2000, and to 344.16 thousand km^2^ in 2010, and then decreased by 326.94 thousand km^2^. The mean soil loss rates changed from 590.66 t·km^−2^·yr^−1^ in 1980 to 634.25 t·km^−2^·yr^−1^ in 2010, and then decreased to 495.66 t·km^−2^·yr^−1^ in 2017. Soil loss rate increased with increasing slope gradient. The changes in SOC loss rates demonstrated the same pattern as that of soil loss, with the largest loss rate of 728.27 kg·km^−2^·yr^−1^. These changes can be explained by changed land use policy and population growth. In future land use management, reasonable implementation of soil conservation measures should be undertaken to reduce soil and SOC losses in the black soil region of northeastern China.

## 1. Introduction

With increasing population and economic development in the world, more land has been converted into arable land [1]. At a global scale, between 1960 and 2015, an average of 2.5 million hectares of land was converted each year into cultivated land [2,3]. Extensification and intensification of agricultural practices during the latter half of the 20th century caused land degradation and poses a serious threat to human health. More than 1.964 billion hectares of land in the world are experiencing different degrees of land degradation [4].

Soil erosion is a pivotal process influencing land degradation [5,6]. Approximately 28 Pg of topsoil was redistributed in the landscape, with approximately 0.5 Pg C per year being mobilized globally [7]. Due to the significant content of C in the soil resource, a small variation in soil C can affect atmospheric CO_2_ concentration [8,9]. Therefore, changes to cultivated land and induced soil erosion and associated SOC dynamics need to be carefully studied [10,11,12].

Land use policy is the driving factor influencing land use change. The number of studies on land use policies’ impacts has increased significantly in recent years. For example, in the United States [13], the adoption of the “Soil Plan” resulted in the deterioration of land conditions, however, subsequent policies, including “the Cropland Conversion Program”, and the “Cropland Adjustment Program”, reversed the situation. In Norway, agricultural subsidy policy in 1950 increased the total cropland area by 80% between 1950 and 1990, resulting in severe soil erosion, whereas subsequent sustainable agricultural production policy alleviated the environmental problems [14]. Similar studies have also been conducted in other regions of the world [15,16].

The black soil in northeastern China has superior physical and chemical characteristics with higher SOC content [6,17]. As the most important commodity grain production area in China, severe soil erosion in the black soil region in China has attracted particular attention [11,18] with many studies on soil erosion and associated SOC loss having been conducted. Based on ^137^Cs technique, Fang et al. [17] described patterns of soil erosion and sediment deposition in a cultivated catchment. Cui et al. [19] evaluated soil erosion changes with increasing slope length using runoff plot data. Li et al. [10] compared the ^137^Cs-derived soil loss rate with SOC and total nitrogen (TN) content, and found that the soil loss rate, SOC and TN content were consistently distributed within a catchment. Wei et al. [18] found that water erosion influenced aggregate size distribution, SOC enrichment, and catchment SOC exports. Zhu et al. [6] also explored SOC dynamics as affected by soil erosion along contrasting hillslopes. In this region, assessments of the impact of soil conservation measures, such as terraces [20], residue cover [21], and rainfall characteristics [22], on soil loss, were also conducted. However, few of these studies were linked to agricultural policies in the black soil region.

Soil erosion modeling is an efficient method for the investigation of the impact of land use change on soil erosion. Among the many models applied, the revised universal soil loss equation (RUSLE) is a simple and plausible empirical physical model that has been used at regional and global scales [23,24]. In our previous study, this model was used to explore land use change impacts on soil erosion at different spatial scales [25]. However, the impacts of cultivated land on soil and SOC losses were not systemically evaluated.

Therefore, the specific aims of this study were to: (i) explore the changes in cultivated land areas during past decades, (ii) identify soil erosion in cultivated land, and (iii) study the impacts of soil erosion on SOC loss for the black soil region of northeastern China. The role of land use policies in influencing land cultivation and associated soil and SOC losses was emphasized in the present study.

## 2. Materials and Methods

### 2.1. Study Area

The study was conducted in northeastern China which includes three provinces (i.e., Heilongjiang, Jilin, and Liaoning Provinces), and the eastern Inner Mongolia Autonomous Region (Figure 1a). It has an area of 1.24 million km^2^, covering three larger rivers, three mountains, and three plains. The Dahinggan Mountains, Xiaohinggan Mountains, and Changbai Mountains are in the northwest, northeast, and southwest of the region. The plains that are situated between the mountains were formed by the piedmont alluvial and flood sedimentation from the Heilong, Songhua, and Wusuli rivers that join in the east of the Sanjiang Plain. The slopes range from zero in the plains to above 60° in the mountainous regions, and the elevations range from −10 to 2666 m (Figure 1a).

The primary land uses in the three provinces are agriculture, forest, and grass (Figure 1c), occupying 41%, 42%, and 12% of the total area, respectively. The major crops are corn and soybean. For the Heilongjiang, Jilin, and Liaoning provinces, their agricultural area percentages for these crops are 65%, 73% and 68%, respectively.

The region studied has a continental climate, with annual precipitation amounts ranging from 242 mm in the northwest to 993 mm in the southeast. The highest mean monthly temperature is 18 °C in July, and the lowest temperature is −20 °C in January. The main soil types are luvisols, phaeozems, chernozems, and gleysols (Figure 1d) which lie on the Quaternary lacustrine and fluvial sand beds or loess sediments.

Fuvisol soils are widely distributed in the mountainous areas, while phaeozems and chernozems, which are called black soils by local people in the Songnen Plain, are mainly distributed in the Heilongjiang and Jilin Provinces. In the present study, the eastern region of the Inner Mongolia Autonomous Region was not included in the study region.

### 2.2. Data Collection

The RUSLE (revised universal soil loss equation) was employed to calculate water erosion. To run this model, several data layers including a digital elevation model (DEM), and data concerning rainfall erosivity, soil erodibility, and land use are required.

The DEM with 90-m resolution was provided through the Geospatial Data Cloud web-site (http://www.gscloud.cn, accessed on 10 August 2020). Soil texture and SOC content data were from the HWSD (the Harmonized World Soil Database) Version 1.2 dataset issued by the Food and Agriculture Organization of the United Nations (FAO-UN; http://www.fao.org, accessed on 10 August 2020). Daily precipitation amounts in 1970–2018 at 120 meteorological stations were acquired from the National Climate Centre of the China Meteorological Administration (http://data.cma.cn/, accessed on 5 July 2020). The land use datasets in 1980 and 2000 were from the Resource and Environmental Sciences, Chinese Academy of Sciences (RES-CAS), and those in 2010 and 2017 were from the GlobeLand30 (http://www.globallandcover.com, accessed on 10 May 2020) and from the research team of Gong Peng in Tsinghua University (http://data.ess.tsinghua.edu.cn/fromglc2017v1.html, accessed on 10 August 2020).

### 2.3. Calculations of Soil Erosion and SOC Loss

#### 2.3.1. Water Erosion

The revised universal soil loss equation (RUSLE) was used to calculate water erosion. It uses five GIS-raster layers, including rainfall erosivity, soil erodibility, slope length and gradient, land use management, and soil conservation practices:(1)$$A=R\cdot K\cdot {\mathrm{LS}}_{2D}\cdot C\cdot P$$
where A is the soil loss rate (t·ha^−1^·yr^−1^), R is the rainfall-runoff erosivity factor (MJ·mm·ha^−1^·h^−1^), K is the soil erodibility factor (t·ha·h·ha^−1^·MJ^−1^·mm^−1^), LS_2D_ is a two-dimension slope length L and slope gradient S factor (−), C is the crop/cover management factor (−), and P is the soil conservation factor (−).

The daily rainfall precipitation amounts in 1970–2018 at different meteorological stations were used to calculate RUSLE-R factor values using the method by Zhang et al. [26], and a co-kriging interpolation was used to derive an R-factor raster data layer. Soil texture and SOC of the top 30 cm soils were available from the HWSD (Figure 1e), and the K factor values were calculated using the EPIC method using soil texture and SOC data. The DEM data was used to calculate the LS_2D_ factor using the method by Desmet and Govers [27]. Detailed calculation methods for R, K, and LS factor values have been described by Fang and Sun [16] and Fang and Fan [25].

The C- and P-factor values were obtained from published papers in northeastern China. They were calculated using runoff plot data [28] or remote sensing data [29]. In the present study, C-factor values for forest, grassland, shrub land, cultivated land, and residential area were set to 0.004, 0.043, 0.07, 0.228, 0.03, and those for bare land, wetland and water bodies, and other unused land, were set to 1, 0, and 0.06, respectively. For the P-factor values, cultivated land and water body were set to 0.352 and 0, and other lands were unit in the study area [25].

#### 2.3.2. SOC Loss

The SOC content in China of the HWSD was determined using a potassium dichromate oxidation with external heating method. Water erosion-induced SOC loss was calculated by multiplying soil loss rate (*A*) and SOC content (g·kg^−1^) in the top 30 cm of soils:(2)$$SOC_{loss}=100A\times SOC$$
where *SOC_loss_* is water erosion-induced *SOC* loss rate (kg·km^−2^·yr^−1^), *A* is calculated by the RUSLE (t·ha^−1^·yr^−1^). Using the HWSD, a top 30-cm soil SOC content layer was obtained and, by applying the calculated soil loss rate by the RUSLE, a spatially distributed SOC loss rate was calculated.

## 3. Results

### 3.1. Changes in Cultivated Land Area

The spatial distributions of cultivated land in the years 1980, 2000, 2010, and 2017 are shown in Figure 2. By visual examination, it can be seen that the cultivated land spreads northward, and that more land was cultivated with time duration. The cultivated land areas increased from 275.11 thousand km^2^ in 1980, to 300.03 thousand km^2^ in 2000, to 344.16 thousand km^2^ in 2010, and then decreased to 326.94 thousand km^2^ (Figure 3a).

Most of the cultivated land was located on the slopes with a gradient of less than 3° and decreased with increasing slope degree (Figure 3b). Temporally, the area percentages of cultivated lands on <3° slopes decreased from 1980 to 2010 and increased from 2010 to 2017 (Figure 3c). However, on steeper slopes with gradients ranging from 3° to 15°, the area percentages of cultivated land presented an opposite trend (Figure 3c).

### 3.2. Changes in Soil Erosion

Spatial distributions of soil erosion are shown in Figure 4. Over time, the mean soil loss rates increased from 590.66 t·km^−2^·yr^−1^ in 1980, to 634.25 t·km^−2^·yr^−1^ in 2010, and then decreased to 495.66 t·km^−2^·yr^−1^ in 2017 (Figure 5a). For individual years, the soil loss rate increased with increasing slope gradient (Figure 5b). Except for slopes in the range 0°–3°, soil loss rate decreased with time duration for each slope range (Figure 5b).

According to the Standards for Classification and Gradation of Soil Erosion (SL 190-2007) [25], around 60% of land experienced soil erosion rates below the tolerable value T in the study region (i.e., 200 t·km^−2^·yr^−1^), 25% land area suffered T–5 T, while around 5%, 3%, and 3% land suffered 5T–10T, 10T–20T, and higher than 20T, respectively (Figure 5c). Temporally, soil erosion rates greater than 5T increased from 1980 to 2010, and then decreased in 2017. In contrast, the areas with soil erosion rate less than one T decreased first and then increased from 2010 onward.

### 3.3. SOC Loss

During 1980–2017, SOC loss rates increased from 702.28 kg·km^−2^·yr^−1^ in 1980, to 710.18 kg·km^−2^·yr^−1^ in 2000, to 726.60 kg·km^−2^·yr^−1^ in 2010, and then decreased to 573.79 kg·km^−2^·yr^−1^ in 2017 (Figure 6a). SOC loss amounts decreased with increasing slope degree. Over 60% of SOC loss occurred on 0°–3° slopes, 20% SOC loss on 3°–5° slopes, 5% SOC loss on 5°–8° slopes, with 3% and 3% SOC losses on 8°–15° and above 15° slopes, respectively. Considering the area (i.e., 788,670 km^2^) of the three provinces, annual SOC loss amounts were 553.87 × 10^3^ t in 1980, 560.10 × 10^3^ t in 2000, 573.05 × 10^3^ t in 2010, and 452.53 × 10^3^ t in 2017.

Annual SOC loss rates increased with increasing slope gradient, from several hundred kg·km^−2^·yr^−1^ on the slopes with gradients 0–3°, to over 10,000 kg·km^−2^·yr^−1^ on the >15° slopes in 1980, 2000, 2010, and 2017. The SOC loss rate in 2010 was the highest on the >3° slopes. Annual SOC loss amount demonstrated an increasing trend with time duration. In 1980, annual SOC loss amounts decreased from 6.956 × 10^4^ t yr^−1^ on the 0–3° slopes, to 3.222 × 10^4^ t·yr^−1^ on the 3°–5° slopes, and then increased to 4.145 × 10^4^ t·yr^−1^ on the 8°–15° slopes. SOC loss amount decreased again on the >15° slopes. A similar pattern of change occurred in other years.

## 4. Discussion

In the study region, the RUSLE and the dataset in the present study were used by Fang and Fan [24], and model performance was verified using published results using runoff plot [30], ^137^Cs technique [19,31], and model simulation methods [32,33]. These confirmed that the model performance was acceptable.

Soil erosion usually increases with increasing slope degree due to a large RUSLE-S factor on steep slopes (Figure 5b). With land conservation, on forest and agricultural lands, SOC content was predicted to be higher on steep slopes, resulting in higher SOC loss rates. However, due to the larger areas on gentle slopes, larger SOC amounts were lost from the gentle slope regions.

In comparison to other land use types, cultivated land usually has a higher soil loss rate. For example, a mean soil loss rate of 404 t·km^−2^·yr^−1^ from agricultural land was reported in the European territory of Russia [34]. In southeastern Nigeria, the mean soil loss rate from the agricultural lands of the upper Eyiobia catchment reached 11,400 t·km^−2^·yr^−1^ [35]. Land use policy usually acts as a land-use change driver [36]. In the study region, the household contract responsibility in 1978 increased peasants’ motivation [37]. Moreover, the grain price began to be protected in 1992—more land implies more grain and increased income. Impacted by these policies, agricultural land spread continuously in 1980–2010 in the mountainous areas at the expense of woodlands, leading to increased soil and SOC loss amounts during 1980–2010. To decrease soil loss, the “Grain to Green Program” policy was introduced in 1999 (Figure 7). This was the largest ecological restoration in the developing world and has made a remarkable contribution to the vegetation recovery in China [38]. In northeastern China, farmland shelterbelts and sand fixation forests were constructed, and many farmlands on steep slopes were converted to forest [33,39]. From 2010 to 2017, forest area increased by 39,928 km^2^, with a change rate of 9%. Similar phenomena also occurred in other countries. In Spain, the EU (European Union) agrarian policy encouraged cultivation on steep slopes and sometimes on new unstable bench terraces, resulting in increased soil erosion [1]. In Italy, the EU’s Common Agricultural Policy (CAP) resulted in reclamation of badlands and degraded grassland for agriculture, and increased the risk of soil erosion [40].

Apart from agricultural policies, the ever-increasing population also acted as an underlying factor that stimulated land use conversions [33]. During past decades, population in the three provinces (i.e., Heilongjiang, Jilin, and Liaoning Provinces) grew progressively from 1960 to 2015. For example, in Heilongjang Province, the population has increased from 18.07 million in 1960 to 37.92 million in 2010 (Figure 8).The increased population led to the establishment of more settlements and clearing of more vegetative cover for food production [16].

Soil erosion can significantly reduce the SOC content in soils [41] and has an adverse effect on land productivity by removing SOC and nutrients [12,42]. During water erosion processes, SOC can be carried away both in dissolved and particulate form. However, the dissolved SOC content in runoff, as well as their enrichment on the sediment [43], was not considered in the present study. Wei et al. [17] found that the SOC enrichment ratio was around 1.6, implying that the SOC loss was underestimated by at least 60%. The declines in the SOC stock in the cultivated lands can thus lead to decreased grain production [6,9,44].

The RUSLE can only estimate water erosion, without considering sediment deposition. In fact, due to gentle and long slopes in the study area, most of the eroded sediment was redeposited on the toe slope or convex areas [17]. This means that the eroded soil and enriched SOC on sediment were redistributed in the landscape. More robust physical models are required to explicate the budget of soil and SOC losses in the cultivated lands.

## 5. Conclusions

The present study evaluated changes to cultivated land and associated soil and SOC losses between 1980 and 2017 using remote sensing data and employing the RUSLE model.

Affected by land use policies and population growth, the soil and SOC loss rates experienced similar changes with increases from 1980 to 2010 and then a decrease in 2017, with a maximum regional soil loss rate of 634.25 t·km^−2^·yr^−1^ in 2010. Because the concept “clear waters and green mountains are as good as mountains of gold and silver” has been emphasized by the Chinese government, current and future policies can be further applied to control soil and SOC losses, though cultivated land area cannot be continuously reduced.

At present, the regional SLR (e.g., 495.66 t·km^−2^·yr^−1^) is still higher than a tolerable rate. Reasonable land use management and best management practices should be implemented which can combat severe soil and SOC losses in the black soil region of northeastern China.

Further study is warranted using physically based soil erosion models to evaluate soil and SOC losses arising from land use changes in the black soil region of northeastern China and other regions in the world.

## Figures and Tables

**Figure 1 ijerph-18-11314-f001:**
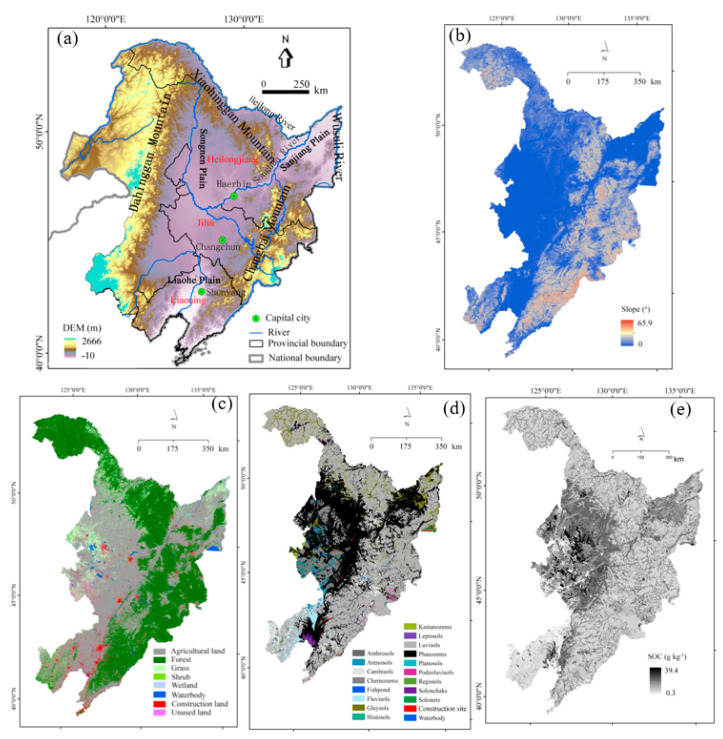
(**a**) Location of the study area, (**b**) slope, (**c**) land use, (**d**) soil distribution, (**e**) soil organic carbon (SOC) content in the top 30 cm soil layer.

**Figure 2 ijerph-18-11314-f002:**
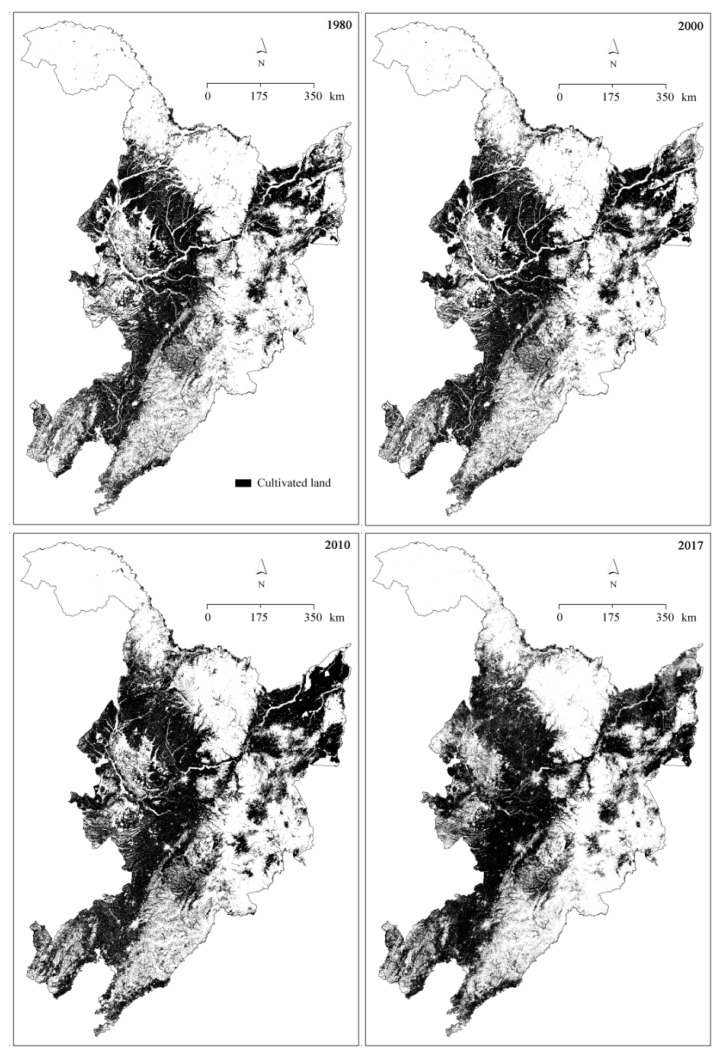
Spatial distribution of cultivated land (black color) in the years 1980, 2000, 2010, and 2017.

**Figure 3 ijerph-18-11314-f003:**
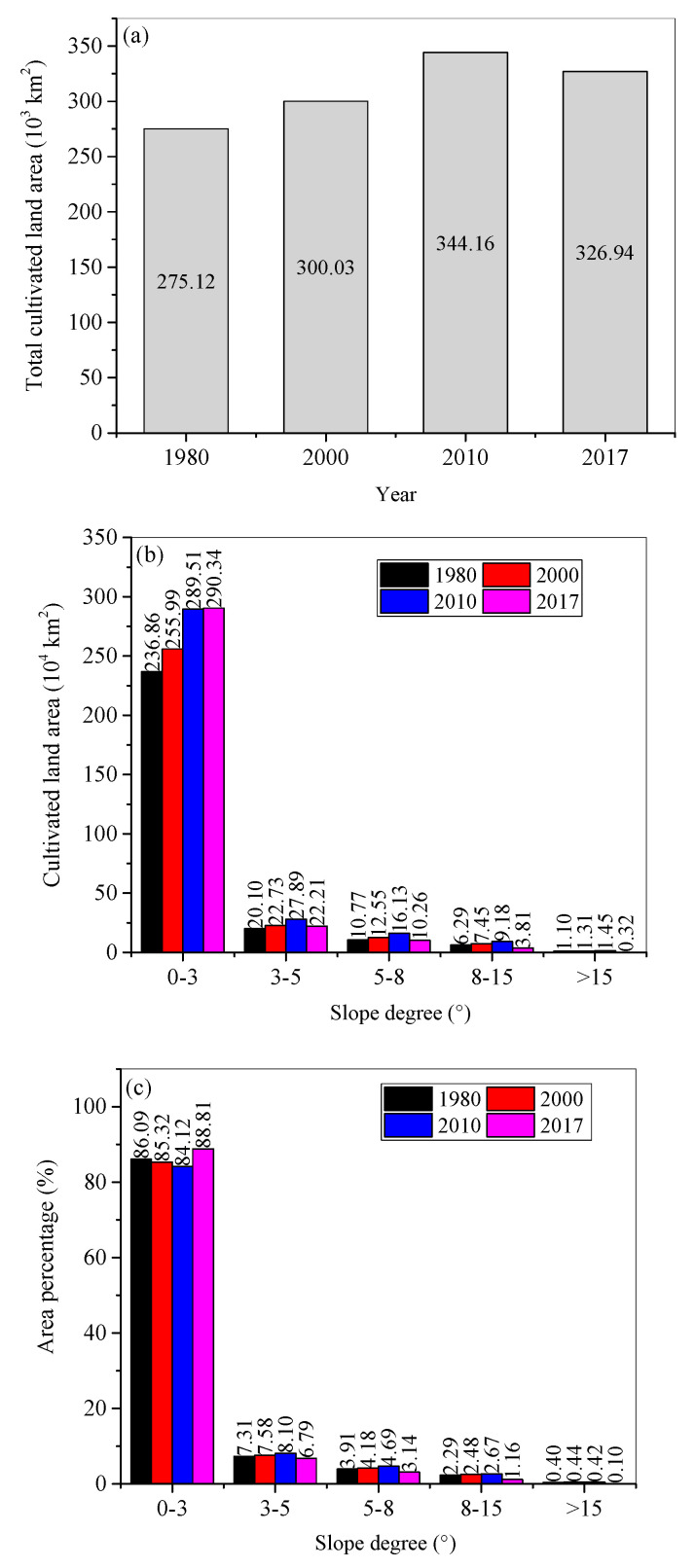
(**a**) Total cultivated land area, and (**b**) cultivated land area and (**c**) area percentages on different slope gradient ranges in the years 1980, 2000, 2010, and 2017.

**Figure 4 ijerph-18-11314-f004:**
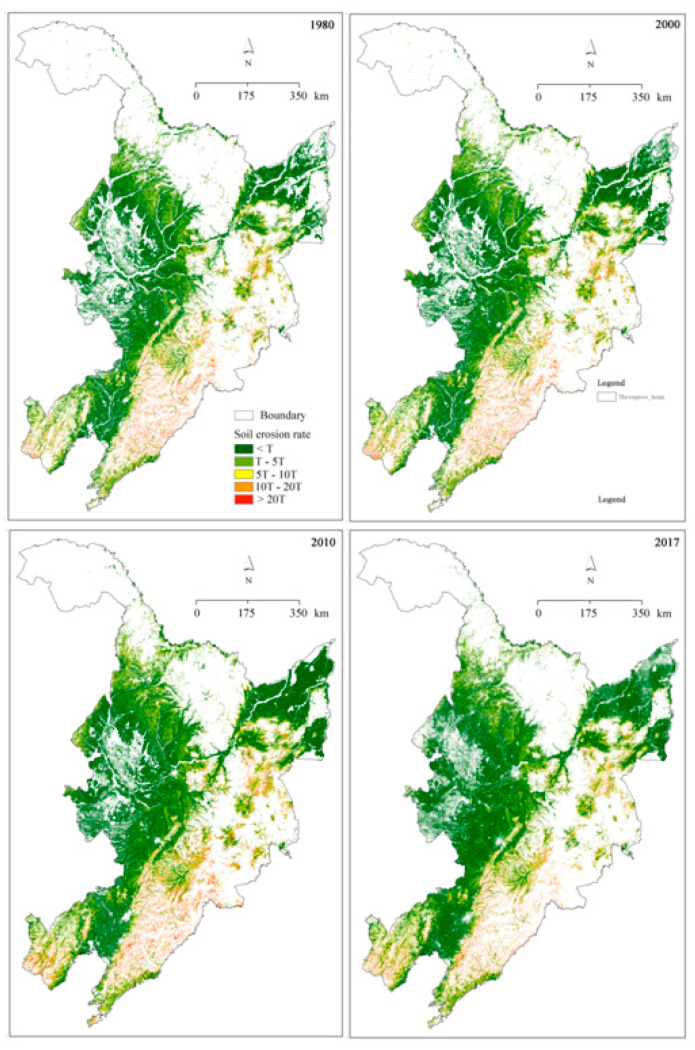
Spatial distributions of soil loss rate on the cultivated slopes in the years 1980, 2000, 2010, and 2017. Note: T = 200 t·km^−2^·yr^−1^.

**Figure 5 ijerph-18-11314-f005:**
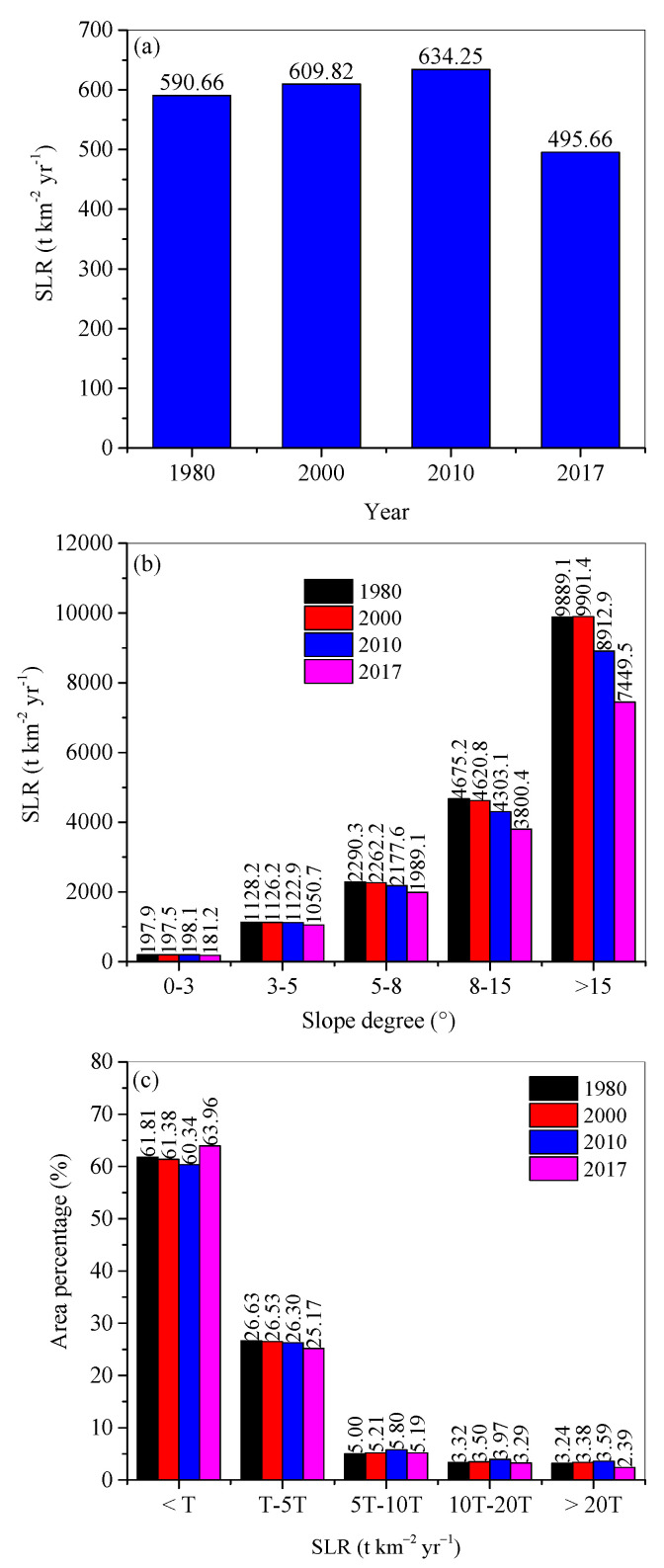
Soil loss rates (SLR) for (**a**) the study region, and (**b**) the areas, and (**c**) their area percentages on different slope ranges in 1980, 2000, 2010, and 2017. Note: T = 200 t·km^−2^·yr^−1^.

**Figure 6 ijerph-18-11314-f006:**
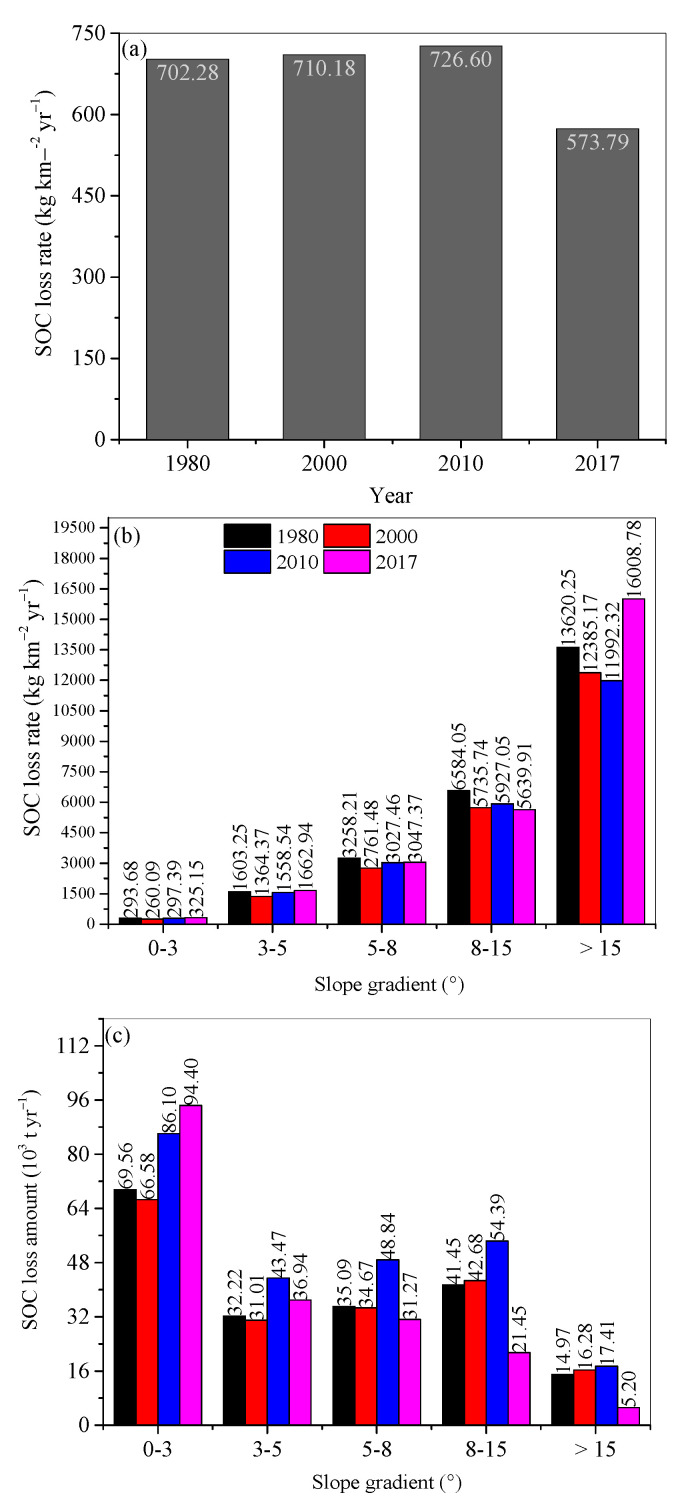
Annual soil organic carbon (SOC) loss rate in (**a**) the study region, (**b**) on different slope ranges, and (**c**) their SOC loss amount in 1980, 2000, 2010, and 2017.

**Figure 7 ijerph-18-11314-f007:**
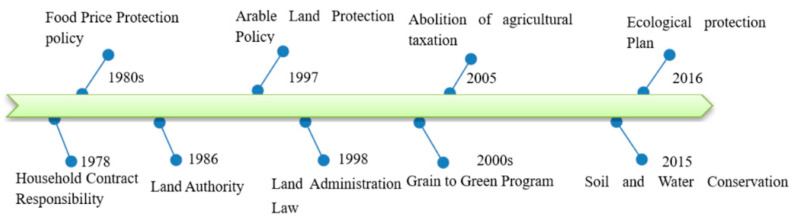
Chronology of the implemented agricultural policies since 1978.

**Figure 8 ijerph-18-11314-f008:**
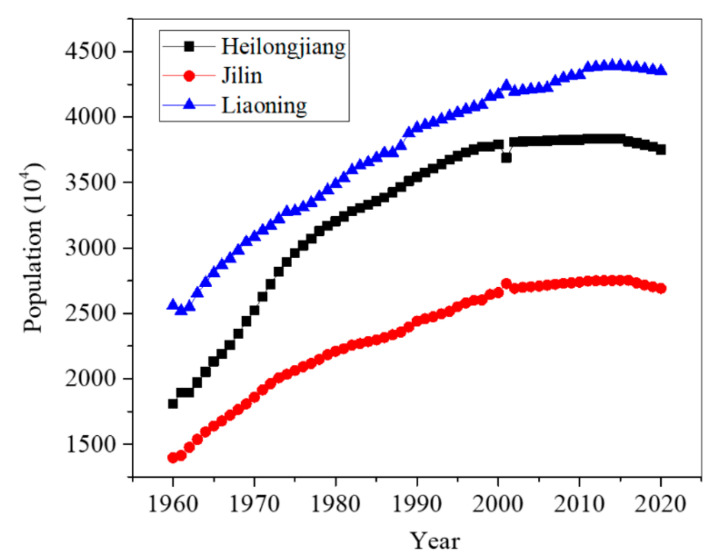
Annual changes of population in Heilongjiang, Jilin, and Liaoning Provinces during 1960–2020.

## Data Availability

The data presented in this study are available on request from the corresponding author.

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
