# Peer review of "Changes in Cultivated Land Area and Associated Soil and SOC Losses in Northeastern China: The Role of Land Use Policies"

_ijerph, 2021, doi:10.3390/ijerph182111314_

Round 1
Reviewer 1 Report
The article " Changes in cultivated land and associated soil and SOC losses in northeastern China: the role of land use policies " presents the results of research aimed at assessing the changes in cultivated land areas during the past decades, soil erosion rate in cultivated land, and the impacts of soil erosion on SOC loss for the black soil region (northeastern China). In the manuscript the authors used the Revised Universal Soil Loss Equation and they integrated remote sensing images in 1980, 1990, 2000, and 2017.
General note.
The concept of the research is very interesting.
Detailed comments:
Line 11 - The abbreviation "SOC" appears in the work for the first time - it should be explained.
Line 105 - there is: (SOC, there should be: SOC
Line 116 - The abbreviation "RUSLE" appears in the work for the first time - it should be explained.
The appearance of Figure 6a should be the same as that of figure 5a
Line 271 - Use italics for the journal name
Line 276 - Use italics for the journal name
Line 323 - Use italics for the journal name
Author Response
Dear reviewer,
Many thanks for your work. According to the suggestions or comments, careful revisions have been done for each comment. All the revised sections or words are labeled in the tracked manuscript.
Line 11 - The abbreviation "SOC" appears in the work for the first time - it should be explained.
Response: Thanks. The full name of SOC was given in the revised manuscript.
Line 105 - there is: (SOC, there should be: SOC
Response: Yes, it was corrected in the revised manuscript.
Line 116 - The abbreviation "RUSLE" appears in the work for the first time - it should be explained.
Response: Thanks. The full name of SOC was given in the revised manuscript.
The appearance of Figure 6a should be the same as that of figure 5a
Response: Thanks.
Line 271 - Use italics for the journal name
Response: Yes, it was corrected in the revised manuscript.
Line 276 - Use italics for the journal name
Response: Yes, it was corrected in the revised manuscript.
Line 323 - Use italics for the journal name
Response: Yes, it was corrected in the revised manuscript.
Reviewer 2 Report
Dear author,
I rate the work highly, but it requires a lot of corrections. I have included all my comments in the file: ijerph-1393218-peer-review-v1 REVIEWED.pdf as reviewer's comments.
The following statements are to be considered:
- Line 14-17 incomprehensible use of the word: "by" If the value decreases from 344.16 thousand square km to 326.94 thousand square km you can say that decreased by 17.22 thousand square km.
You can find the same term in several places in the manuscript i.e. lines 146-150, 177-180, 254-256
- Line 49, here you need to specify in advance what region you are talking about, indicate the map (figure)
- Line 132, here the author should provide a method (see manuscript)
- Line 139, similar to before, there is no information about SOC content determination (method)
- Line 152, cultivated land is of course marked as a black colour
- Line 153, Inconsistent information appears in Figure 4. The Sum of specific areas (b) is not equal with total cultivated land areas (a)
- Line 171, How was the tolerable T-value determined (estimated)? Where did this value come from, please provide the source of data in the references section.
- Line 182, How to understand your calculation? SLR in 1980 was 590.66 km2. If summarize soil losses according to slope degrees the result for 1980 is 18180.7 km2, not 590.66 - how to explain the difference? For 2000, 2010 and 2017 the same situation?
- Line 183, Wherein this manuscript does the author discuss soil loss rates?
- Line 251, Please rewrite the conclusions section. To the conclusions can not be copied sentences from the abstract. The author initially provided this information in the Abstract section. Here the author should briefly write the direction of the changes and what the author thinks the consequences will be related to the long term changes resulting from both soil organic carbon losses and soil losses. This study emphasizes the role of land use policies, but it should also be indicated what directions in this respect will be future-proof.
The rest of my suggestions you can find in the manuscript.pdf file
I hope that my comments will be understandable and will help the author to properly prepare the manuscript, making it more readable
Thank you
Reviewer

Author Response
Dear reviewer,
Many thanks for your work. According to the suggestions or comments, careful revisions have been done for each comment. All the revised sections or words are labeled in the tracked manuscript.
1, Please delete the highlighted text
Response: “Corresponding author. Email address” was deleted.
2, line 14-17 incomprehensible use of the word: "by" If the value decreases from 344.16 thousand square km to 326.94 thousand square km you can say that decreased by 17.22 thousand square km. You can find the same term in several places in the manuscript i.e. lines 146-150, 177-180, 254-256
Response: Thanks. All these were corrected in the revised manuscript.
3, could you cite following article: Sądej W., Żołnowski A.C. (2019): Comparison of the effect of various long-term fertilization systems on the content and fractional composition of humic compounds in Lessive soil. Plant Soil Environ., 65: 172–180. https://doi.org/10.17221/777/2018-PSE
Response: Yes, the paper was cited in the revised manuscript.
4,- Line 49, here you need to specify in advance what region you are talking about, indicate the map (figure)
Response: Yes, the region was given. However, figure 1 was not cited here because it is given in the 2. Materials and methods section.
5, - Line 132, here the author should provide a method (see manuscript)
Response: Yes. In the study region, C values were given by runoff plots and/or remote sensing methods. This information was given and two references were also provided here in the revised manuscript.
6, There is no information about the SOC content (g kg-1) determination method here. Please provide this information (method and apparatus)
Response: Thanks. The method to determined SOC content was added in the revised manuscript.
7, Some suggestions for figure 3: Total cultivated land area, 202.4+42.1+13.2+11.8+4=273.5, 275.12-273.5=1.62, where is 1.62 thousands square kilometers? 300.03-298,2=1.83? 344.16-342.2=1.96? 326.94-324.8=2.14? Could you explain?
Response: Thanks. “Area” in Figure 3a was replaced by “Total cultivated land area”, and the area and area percentage for each slope range was updated.
8, How was the tolerable T-value determined (estimated)? Where did this value come from, please provide the source in the references section.
Response: Yes. The tolerable T was determined by the Standards for Classification and Gradation of Soil Erosion (SL 190-2007). A reference was cited here to provide the data source.
9, In the text, the author do not refer to figure 5a Soil loss rates (SLR) for the study region.
Response: It was added in the 3.2 section.
10, Figure 5: Percentage values seems to be correct, but please check it again.
Response: The values were checked.
11, Figure 6: Try recalculate these data again
Response: Thanks. They were recalculated. Accordingly, the corresponding content in the main text was also updated for the updated figure data.
12, - Line 152, cultivated land is of course marked as a black colour
Response: Yes, an explanation was added in the caption.
13, How to understand your calculation? SLR in 1980 was 590.66 km2. If summarize soil losses according to slope degrees the result for 1980 is 18180.7 km2, not 590.66 - how to explain difference? For 2000, 2010 and 2017 the same situation?
Response: They should not be summarized. The 590.66 is the mean value for the study area. In contrast, Figure 5b showed soil loss rate at different slope. The mean value of 590.66 should equal soil loss rate at different slope range multiplying the area percent of each slope range in the study area.
- For Figure 6, try recalculate these data again.
Response: Yes. They were recalculated in the revised manuscript, and the data were updated in the revised manuscript.
15, - Line 183, Wherein this manuscript does the author discuss soil loss rates?
Response: They were discussed in 3.2 section, and more information was updated in the revised manuscript.
16, - Line 251, Please rewrite the conclusions section. To the conclusions cannot be copied sentences from the abstract. The author initially provided this information in the Abstract section. Here the author should briefly write the direction of the changes and what the author thinks the consequences will be related to the long term changes resulting from both soil organic carbon losses and soil losses. This study emphasizes the role of land use policies, but it should also be indicated what directions in this respect will be future-proof.
Response: According to the valuable suggestions in the comments, the conclusion was rewritten, including the change direction of soil and SOC loss, the consequences, and the development direction of policies for soil and SOC losses control.
Round 2
Reviewer 2 Report
Dear Author,
All comments of the reviewer suggested in the first review were correctly interpreted and included in the text of the manuscript. There are a few small details that are more of an editorial nature and need to be included. Congratulations, good job!
Regards
Reviewer
